# In-Water and On-Land Swimmers’ Symmetry and Force Production

**DOI:** 10.3390/ijerph16245018

**Published:** 2019-12-10

**Authors:** Diogo D. Carvalho, Susana Soares, Rodrigo Zacca, Daniel A. Marinho, António J. Silva, David B. Pyne, J. Paulo Vilas-Boas, Ricardo J. Fernandes

**Affiliations:** 1Centre of Research, Education, Innovation and Intervention in Sport, Faculty of Sport, University of Porto, 4200-450 Porto, Portugal; diogoduarte_03@hotmail.com (D.D.C.); susana@fade.up.pt (S.S.); rodrigozacca@yahoo.com.br (R.Z.); jpvb@fade.up.pt (J.P.V.-B.); 2Porto Biomechanics Laboratory, University of Porto, 4200-450 Porto, Portugal; 3Department of Sport Sciences, University of Beira Interior, 6201-001 Covilhã, Portugal; marinho.d@gmail.com; 4Research Center in Sport, Health and Human Development, CIDESD, 5001-801 Vila Real, Portugal; ajsilva@utad.pt; 5Department of Sport Sciences, University of Trás-os-Montes and Alto Douro, 5000-801 Vila Real, Portugal; 6Research Institute for Sport and Exercise, University of Canberra, ACT, Canberra 2601, Australia; david.Pyne@canberra.edu.au

**Keywords:** biomechanics, isokinetic, tethered swimming, asymmetry, swimming

## Abstract

Although performance and biomechanical evaluations are becoming more swimming-specific, dryland testing permits monitoring of a larger number of performance-related variables. However, as the degree of comparability of measurements conducted in-water and on land conditions is unclear, we aimed to assess the differences between force production in these two different conditions. Twelve elite swimmers performed a 30 s tethered swimming test and four isokinetic tests (shoulder and knee extension at 90 and 300°/s) to assess peak force, peak and average torque, and power symmetry index. We observed contralateral symmetry in all the tests performed, e.g., for 30 s tethered swimming and peak torque shoulder extension at 90°/s: 178 ± 50 vs. 183 ± 56 N (*p =* 0.38) and 95 ± 37 vs. 94 *±* 35 N × m (*p =* 0.52). Moderate to very large direct relationships were evident between dryland testing and swimming force production (r = 0.62 to 0.96; *p* < 0.05). Swimmers maintained similar symmetry index values independently of the testing conditions (r = −0.06 to −0.41 and 0.04 to 0.44; *p =* 0.18–0.88). Asymmetries in water seems to be more related to technical constraints than muscular imbalances, but swimmers that displayed higher propulsive forces were the ones with greater force values on land. Thus, tethered swimming and isokinetic evaluations are useful for assessing muscular imbalances regarding propulsive force production and technical asymmetries.

## 1. Introduction

Biomechanical characteristics are important swimming performance determinants, fundamental in understanding propulsion mechanics in the highly specific hydrodynamic environment. Analyzing swimmers’ force production should be a priority in their training control and research, as an effective propulsion is fundamental for competitive success [1,2,3]. For this purpose, tethered swimming has been one of the most frequently used methods, yielding substantial associations between tethered forces and swimming performance in sprint events [4,5,6]. These studies underpin the notion that movements used in training and testing should be mechanically similar to those used in competition [7].

Complementarily, it is known that similar force application from both right and left body sides can positively affect swimming performance, particularly by reducing drag, promoting better body alignment and lowering intracycle velocity variations [8]. In fact, ~96% of the human population present a perceptible asymmetry level [9], which may negatively influence exercise performance, mainly in cyclic and continuous sports [10,11]. Since elite swimmers spend a huge number of hours in-water and land workouts, they are more vulnerable to overuse injuries, particularly if they present muscular and technical imbalances [12]. Furthermore, if hydrodynamic and inertial limitations are also considered, swimmers need to be as symmetrical as possible when propelling through the water.

The isokinetic dynamometer is the laboratorial gold standard to evaluate asymmetries and imbalances in force production, permitting analysis of specific muscle groups and movements [12,13,14]. However, even if its use is common in muscular symmetry analysis, it does not allow evaluating specific swimming movements. For this reason, tethered swimming has been conducted to quantify and analyze both force production and asymmetries during swimming [11,15,16]. Surprisingly, it is unclear if measurements obtained from tethered swimming and isokinetic dynamometer are related, and whether swimmers generate a similar level of force production symmetry in-water and dryland conditions.

Using an innovative approach, with closer approximation to real-world swimming characteristics, of assessing shoulder and knee extension force values (hand placement, angular velocities, isolated and combined analysis), the aim of the current study was to determine the degree of association between tethered swimming peak propulsive forces and dry land isokinetic torque and power. We also sought to determine whether there is a similar contralateral symmetry regarding force production in both the above referred conditions and the nature of their symmetry index (SI) relationships. Finally, we determined whether swimming symmetry can be identified by using only the first 10 upper limb cycles during tethered swimming.

## 2. Materials and Methods

### 2.1. Participants

Twelve elite swimmers (seven males and five females) voluntarily participated in the current study, presenting the following physical and competitive characteristics (respectively): 20.9 *±* 3.4 vs. 19.0 ± 2.2 years of age, 77.1 ± 8.9 vs. 59.6 ± 5.0 kg of body mass, 1.82 ± 0.04 vs. 1.69 ± 0.06 m of height, 23.3 ± 2.1 vs. 20.8 ± 1.8 of body mass index and 841 ± 37 vs. 816 ± 38 Fédération Internationale De Natation points of best competitive performance. Eight swimmers presented right lateral preference and four were left-handed. The experimental moment comprised a tethered swimming test in an indoor 25 m pool and four laboratory isokinetic evaluations, performed in randomized order on consecutive days. Subjects were familiarized with the two apparatus, informed about the purpose of the evaluations and any known risks, providing individual consent for participation in accordance with the local research ethics committee (CEFAD 28 2019) and Declaration of Helsinki.

### 2.2. Design and Procedures

After a 1000 m low-to-moderate intensity warm-up, swimmers performed a 30 s maximal front crawl (*n* = 6) and backstroke (*n* = 6) tethered swimming test using a belt attached to a steel cable (5 m length and 5.7° angle with the water surface) [17]. Its beginning and end were indicated by an acoustic signal, with swimmers adopting a horizontal position with the cable fully extended just before the starting signal. The data collection only commenced after the first upper limbs cycle was completed. A load-cell connected to a Globus Ergometer data acquisition system (Globus, Codognè, Italy) obtained force measurements at a rate of 100 Hz [5]. Tests were recorded using one surface video camera (50 Hz, Sony^®^ Handycam HDR-CX160E, Tokyo, Japan) placed in the sagittal plane, with video acquisition synchronized with force data to better identify the beginning and end of each upper limb cycle.

In isokinetic dynamometer (Biodex System 4-Biodex Medical Systems, Inc., Carrollton, TX, USA), after freely warmed-up without constraint of velocity, swimmers performed a shoulder and knee extension/flexion (representing the front crawl and backstroke upper limb action with 180° range of motion and leg movement during the lower limb actions with 90° range of motion, respectively) at 90 and 300°/s and a sampling rate of 100 Hz. These angular velocities represented a low (used for construct ratios of forces in a clinical point-of-view) and a high (similar to competitive swimming). Swimmers were positioned in the dynamometer chair with a forearm supination similar to hand positioning during swimming (Figure 1). The tests consisted in five sub-maximal repetitions at each pre-defined angular velocity, 10 maximal repetitions with 90 s and 10 min of rest permitted between side trials and movements, respectively. Swimmers were instructed to generate maximal one-way force during the phase of knee and shoulder extension to form a more realistic simulation of the aquatic environment motion.

### 2.3. Measures

Tethered swimming peak forces of each upper limb action were extracted from individual force vs. time curves synchronized with the video recording to associate each peak force value to one upper limb cycle body side [16], involving a two-stage process; (i) characterizing the initial 10 upper limb cycles, i.e., the time duration associated with changings in energetic system dominance and muscle recruitment [18,19]; (ii) characterizing all upper limbs cycles performed during the entire 30 s, representing the gold standard test to quantify the anaerobic capacity, the Wingate test [16,20]. During each isokinetic evaluation, the peak torque (the highest value observed in the 10 repetitions), mean peak torque (mean of the 10 peaks obtained) and mean power values were determined. Shoulder and knee isokinetic force variables were analyzed individually and in combination to represent the group muscles used during front crawl and backstroke swimming.

SI was calculated for all force variables according to the following equation:SI (%) = (preferred − nonpreferred) × [0.5 × (preferred − nonpreferred)] ^ (−1) × 100(1)
in which values between −10 < SI < 10% and −10 > SI > 10% indicate symmetry and asymmetry, respectively [21]. SI values were made positive, obtaining means without a zero-central tendency to compute absolute asymmetries between tethered swimming and isokinetic dry land testing. This approach permitted comparisons of overall level of asymmetry values irrespective of the preferred side.

### 2.4. Statistical Analysis

Descriptive analyses (mean and standard deviation) were obtained for all variables and data checked for distribution normality with the Shapiro-Wilk test. Spearman correlation coefficients were calculated between isokinetic and tethered swimming force related variables and correspond SI. Intraclass correlation coefficients (ICC) were assessed to quantify the degree of reliability between repetitions of each body side in isokinetic and tethered swimming. Correlations were interpreted as follows [22]: 0–0.25 (little), 0.26–0.49 (weak), 0.50–0.69 (moderate), 0.70–0.89 (strong) and 0.90–1.0 (very strong). The Wilcoxon signed-rank test was used to compare the forces produced by the preferred and the non-preferred body side and the mean absolute SI obtained with isokinetic and tethered swimming tests.

Effect size was calculated using the ratio of z score and square root of the number of observations, and interpreted as 0–0.09 (trivial), 0.1–0.29 (small), 0.3–0.49 (medium), 0.5–0.69 (large) and ≥0.7 (very large) [23]. A Bland-Altman plot analysis [24] was also performed to characterize the differences between SI of tethered swimming and isokinetic evaluations, and between SI of the first 10 upper limb cycles and 30 s in the tethered swimming (the last one used as reference method). A *p* ≤ 0.05 level of significance was accepted.

## 3. Results

Tethered swimming and force production isokinetic variables were moderately to very largely directly related, with correlation values ranging from r = 0.62 to 0.96. The peak torque, mean torque, mean power and mean peak forces values of the first 10 upper limb cycles and 30 s of front crawl and backstroke tethered swimming, for both preferred and nonpreferred body sides, are presented in Table 1. When analyzed as combined isokinetic forces, the relationships ranged from r = 0.65 to 0.94. In most of the evaluated variables, the non-preferred side presented higher absolute correlations than the preferred side. Values obtained during the first 10 s of tethered swimming presented similar correlation to the isokinetic forces when compared to the 30 s data.

No substantial differences were evident between SI of isokinetic and tethered swimming force production or between first 10 upper limb cycles and 30 s of tethered swimming (Figure 2). Only in two swimmers, an asymmetrical force production in tethered swimming (SI > 10%) was identified. However, in the isokinetic evaluations, eight swimmers presented asymmetrical force production (at least in one of the studied variables), reduced to three swimmers when the shoulder and knee extension combined variables were evaluated. Absolute values of SI obtained from tethered swimming and isokinetic force variables showed no substantial differences, but a high variability was evident between subjects (Figure 2). The correlation analyses showed that only the SI of 10 upper limb cycles had a moderate inverse relationship with SI of power of knee extension at 300°/s (r = −0.66) and combined average power SI (r = −0.60). No substantial relationships in the other SI (r = −0.06 to −0.41 and 0.04 to 0.44 with *p* = 0.18 to 0.88).

Swimmers showed a similar force production between preferred and nonpreferred body sides (both in tethered front crawl and backstroke swimming) and isokinetic tests (Table 2; *p* = 0.18 to 0.97). In the tethered swimming and isokinetic tests, intraclass correlation between repetitions for both preferred and nonpreferred side were excellent (ICC: 0.989 to 0.997; 95% CI: 0.963 to 0.999 with *p* < 0.001). Agreement analyses between SI’s of isokinetic evaluations and 30 s tethered swimming presented a bias standard deviation >9% and the 95% limits of agreement higher than 19%, demonstrating that the SI were independent from each other (see Table 2). The SI of the first 10 upper limb cycles and 30 s in tethered swimming test showed a very strong relationship (r = 0.99). In addition, comparison between the SI calculated with the first 10 upper limb cycles and 30 s of tethered swimming showed a good agreement with a bias of −0.7 ± 1.2% and 95% limits of agreement of −3.0 and 1.6%, resulting in a consistent behavior of SI during the 30 s effort.

## 4. Discussion

Our results indicate moderate to very strong relationships between in-water and dryland conditions force production, which supports the first hypothesis that forces produced in land condition are closely related with the capability to apply force in-water. We also identified similar force production between preferred and nonpreferred body sides, showing that elite swimmers, as a consequence of their training, exhibit upper and lower body symmetry. Our swimmers presented similar but independent contralateral symmetry during swimming and dryland testing. The very strong relationship between isokinetic and tethered forces supports the assertion that isokinetic ones are related to swimming performance and reinforced that the 30 s tethered swimming test is a good predictor of swimming performance [6,17].

The above outcomes are in accordance with the study that reported an inverse relationship between isokinetic knee extension force and mono-fin swimming 100 m time [25]. However, some studies indicated that isokinetic force variables were not associated with swimming performance [26,27,28]. These last studies used dryland tests that employed different movements types than those used in swimming (different muscular groups and patterns). Regarding not only isokinetic variables, other studies showed strong relationships (e.g., r = 0.78–0.94) between force variables in dryland exercises (bench press and squad jumps) and force produced during tethered swimming [29,30]. Taken together, these data reinforce the notion that forces produced in dryland testing can be transferred to water when patterns of movement angular velocities and amplitude are similar.

The biomechanical asymmetries are useful both for clinical and research purposes, permitting the characterization of the functional imbalances between contralateral limbs [31]. Theoretically, the front crawl and backstroke intracycle velocity variation can be enhanced by this functional imbalance, increasing the energy cost of swimming [32,33] and deteriorating body postures with increasing hydrodynamic drag [10] at a given swimming speed. The current results demonstrate that elite swimmers are more symmetrical than lower level swimmers (data from [16]), since we did not observe considerable differences between the force production of contralateral body sides.

However, the mean values of the SI need to be considered with caution, as asymmetry measurements are direction-dependent and a tendency for both preferred and non-preferred predominance will make the mean values tend towards zero. Therefore, comparing absolute values between conditions (in-water and dryland) is preferred. The similar SI in-water and in-dry land conditions observed in this study likely reflects the technical training to build a balanced and efficient swimming technique. This process typically involves a combination of pool-based dryland training enhance bilateral symmetry and muscular proficiency.

Despite similar tethered and isokinetic force production SI, the correlation analyses show the few substantial relationships between almost all tethered and isokinetic SI. The agreement analyses presented a bias of ~10 and 95% limits of agreement ~20% indicating why SI values were not underpinning the change of SI direction in almost all swimmers evaluated. These data indicate that during swimming, one upper limb is typically used for propulsion and the other for control and support. In addition, it appears that the preferred upper limb is the limb of control and support during swimming, while the non-preferred limb is used mainly for propulsion. This pattern of limb control and propulsion explains the higher correlation values on the non-preferred body side.

This outcome highlights that the explanations of asymmetry observed in-water are more related to swimming technique adaptations consequent of the aquatic environment and to a lesser extent to muscular imbalances [34]. We expected changes on SI across 30 s test in response to the change of energetic system and muscular recruitment [18,19]. However, when examined the force (a)symmetry development along the maximal test, very strong direct relationships (r = 0.98) and similar values between the SI from the first 10 upper limb cycles and 30 s where found. The behavior of SI during the first 10 cycles showed high agreement with the 30 s data, enhancing the capability of 10 cycles to indicate likely patters during the overall effort. These results can be justified by the adaptations of swimming technique during the test being more important than the capacity of force production for SI behavior. However, this outcome is contrary to that shown in a previous study [16] using only front crawl tethered swimming in low-level swimmers. These outcomes indicate that high-level swimmers can maintain symmetry during the effort even when the absolute force produced is diminishing.

## 5. Conclusions

There is a direct relationship between forces produced in isokinetic and tethered swimming conditions for both front crawl and backstroke. We also confirmed similar force production symmetry in these swimming techniques. Coaches and swimmers may use isokinetic and tethered swimming evaluations to determine muscular and swimming technical imbalances, and track training-induced changes in these measures within and between swimming seasons. The first 10 cycles in tethered swimming should be sufficient to define the behavior/patterns of the symmetry index during an all-out testing.

## Figures and Tables

**Figure 1 ijerph-16-05018-f001:**
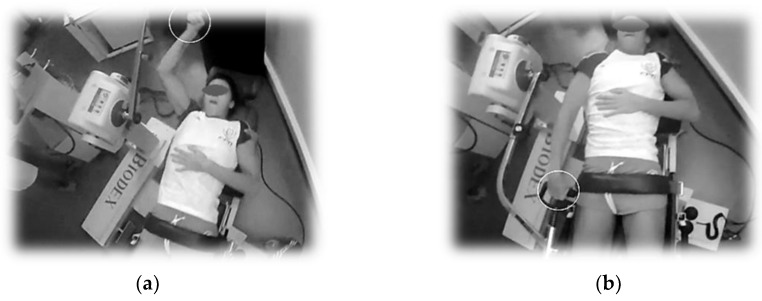
Initial and final positioning of the hand (white circle) during upper limb simulated action isokinetic test ((**a**) and (**b**) panels, respectively).

**Figure 2 ijerph-16-05018-f002:**
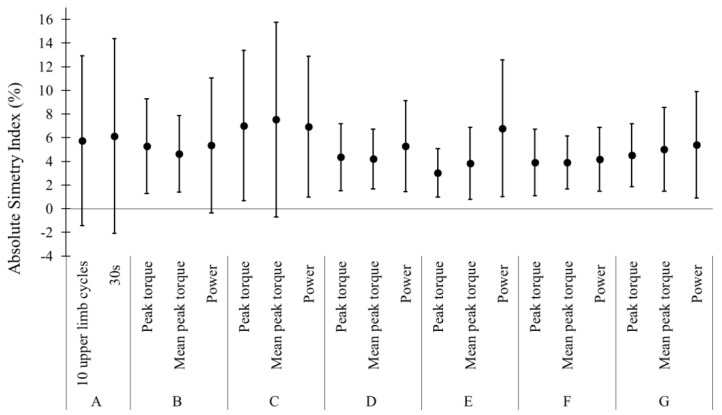
Mean and SD values of absolute symmetry index of tethered swimming (A), upper limb simulated action (90°/s and 300°/s: B and C), knee extension (90°/s and 300°/s: D and E), combined upper limb simulated action and knee extension (90°/s and 300°/s: F and G).

**Table 1 ijerph-16-05018-t001:** Correlation coefficients between tethered swimming and isokinetic variables for preferred and non-preferred upper body sides. All associations were significant (*p* < 0.05).

Isokinetic Variables	Front Crawl and Backstroke Tethered Swimming			
	10 Upper Limb Actions		30 s	
	Preferred	Nonpreferred	Preferred	Nonpreferred
Upper limb simulated action				
Peak torque 90°/s	0.83	0.92	0.82	0.90
Mean peak torque 90°/s	0.80	0.94	0.79	0.91
Mean power 90°/s	0.80	0.96	0.79	0.93
Peak torque 300°/s	0.62	0.67	0.66	0.66
Mean torque 300°/s	0.64	0.72	0.68	0.71
Mean power 300°/s	0.81	0.94	0.83	0.92
Knee extension				
Peak torque 90°/s	0.68	0.90	0.68	0.81
Mean peak torque 90°/s	0.80	0.90	0.78	0.84
Mean power 90°/s	0.79	0.91	0.78	0.85
Peak torque 300°/s	0.73	0.92	0.71	0.86
Mean torque 300°/s	0.72	0.93	0.69	0.89
Mean power 300°/s	0.71	0.96	0.69	0.94
Combined				
Peak torque 90°/s	0.79	0.91	0.78	0.85
Peak torque 300°/s	0.69	0.84	0.72	0.84
Mean peak torque 90°/s	0.79	0.94	0.78	0.89
Mean peak torque 300°/s	0.65	0.87	0.69	0.87
Mean power 90°/s	0.83	0.92	0.83	0.87
Mean power 300°/s	0.76	0.83	0.72	0.85

**Table 2 ijerph-16-05018-t002:** Isokinetic and tethered swimming variables mean ±SD for preferred and nonpreferred body side and effect size. Bias and 95% limits of agreement of estimations between symmetry indexes of isokinetic evaluations and 30 s tethered swimming.

Variables	Preferred	Nonpreferred	Effect Size	Bias	95% CI
Upper limb simulated action					
Peak torque 90°/s (N × m)	95.10 ± 36.96	93.89 ± 35.14	−0.14	1.5 ± 13.4	−25–28
Mean peak torque 90°/s ( N × m)	87.78 ± 33.92	86.19 ±31.72	−0.15	1.9 ± 10.8	−19–23
Mean power 90°/s (W)	99.72 ± 38.46	98.47 ± 36.89	−0.13	−2.0 ± 9.8	−17–21
Peak torque 300°/s (N × m)	100.92 ± 23.76	105.22 ± 22.93	−0.28	−1.8 ± 16.9	−35–31
Mean peak Torque 300°/s ( N × m)	93.63 ± 23.46	95.64 ± 20.50	−0.03	−0.6 ± 18.5	−37–36
Mean power 300°/s (W)	186.55 ± 90.21	192.69 ± 86.71	−0.11	−1.6 ± 12.7	−26–23
Knee extension					
Peak torque 90°/s (N × m)	185.60 ± 50.06	186.86 ± 50.30	−0.08	0.6 ± 10.7	−20–22
Mean peak torque 90°/s ( N × m)	171.25 ± 47.09	170.72 ± 44.29	−0.03	1.0 ± 10.5	−20–22
Mean power 90°/s (W)	173.51 ± 51.67	177.41 ± 47.48	−0.16	−0.6 ± 10.6	−21–20
Peak torque 300°/s (N × m)	124.36 ± 39.95	127.34 ± 38.57	−0.27	−0.9 ± 11.8	−24–22
Mean peak torque 300°/s ( N × m)	114.33 ± 36.98	116.48 ± 35.02	−0.18	−0.7 ± 12.7	−26–24
Mean power 300°/s (W)	306.94 ± 110.05	307.93 ± 91.57	−0.02	−0.1 ± 15.0	−29–29
Combined					
Peak torque at 90°/s (N × m)	280.70 ± 85.52	280.75 ± 83.84	−0.02	0.9 ± 11.3	−21–23
Peak torque 300°/s (N × m)	225.28 ± 61.35	232.56 ± 58.45	−0.30	−1.3 ± 13.7	−28–26
Mean peak torque 90°/s ( N × m)	259.03 ± 79.99	256.91 ± 75.01	−0.03	−0.6 ± 14.7	−29–28
Mean peak torque 300°/s ( N × m)	207.97 ± 57.93	212.12 ± 53.12	−0.16	1.3 ± 10.4	−19–22
Mean power 90°/s (W)	273.23 ± 87.72	275.88 ± 82.97	−0.10	0.2 ± 9.7	−19–19
Mean power 300°/s (W)	493.79 ± 197.05	500.62 ± 175.73	−0.05	−0.8 ± 13.0	−26–25
Tethered swimming					
10 upper limb actions (N)	183.20 ± 49.41	186.68 ± 56.17	−0.10	-	-
30 s (N)	177.71 ± 49.55	183.44 ± 56.31	−0.19	-	-

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
