# Peer review of "In-Water and On-Land Swimmers’ Symmetry and Force Production"

_ijerph, 2019, doi:10.3390/ijerph16245018_

Round 1

Reviewer 1 Report

General comments:

The article “In-water and on land swimmer’s symmetry and force 2 production”  include  interesting data, which might be used in sport practice, however the novelty of the study is should be more justified in introduction. On the other hand, the article lack in description of methodology, which is the main weakness of this paper. Thus, I highly recommend to include more details about data collection of peak forces during tethered swimming. The discussion is appropriate and includes sufficient generalization.  

Specific comments:

Line 22 -23 : “ isokinetic laboratory evaluations “ is not a right term use “isokinetic test of…..”

Line 26: The  N·m does not need the dot between standard is Nm.

Line 24 26: This results are not clear side is reported value.

Line 37: instead of “ resistive water environment” you should write specific hydrodynamic environment.

Line 38 – 40: If there are already studies showing  associations between tethered forces and swimming performance in sprint events, why you are performing your research. Add some statement what is not done already here.

Line 65 – 66: the additional aim should be included in abstract as well.

Line 72: define abbreviation.

Line 94: what was the range of motion for knee extension and shoulder as well. Specifically, did you focused on certain specific angle during the testing protocol.

Line 80 and 106.: There is lack of description how peak forces during tethered swimming  were calculated - measured. The  reference should not substitute the explicit statement how you obtained data.

Line 110: I don’t get why you mentioning Wingate test.

Line 112: Formatting

Line 123: Please report also ICC between isokinetic and swimming stride repetitions.

Reviewer 2 Report

Authors are acknowledged by the quality of the present manuscript although they should take the following points into consideration: 

first two lines of introduction...are they really needed? line 60:  according to information in the introduction section the present reviewer does not understand why this is an innovative approach. please explain. lines 90-92 please rephrase as it seems to be some misspelling in line 90. The present reviewer does not understand the differences between the two first sentences of discussion. aren't both sentences referring to the same idea? line 191 does a comma seem to be missing after "movement"? conclusions: according to the aims of the present study indicated at the end of introduction, there should be any conclusion about the 30s to 10s differences...

Round 2

Reviewer 1 Report

The authors addressed all of my comments and the manuscript is significantly improved, therefore I recommend this article for publication.